# Liposomal Encapsulation of Carob (*Ceratonia siliqua* L.) Pulp Extract: Design, Characterization, and Controlled Release Assessment [note 1]

**DOI:** 10.3390/pharmaceutics17060776

**Published:** 2025-06-13

**Authors:** Aleksandra A. Jovanović, Dragana Dekanski, Milena D. Milošević, Ninoslav Mitić, Aleksandar Rašković, Nikola Martić, Andrea Pirković

**Affiliations:** 1Institute for the Application of Nuclear Energy INEP, University of Belgrade, 11080 Belgrade, Serbia; ajovanovic@inep.co.rs (A.A.J.); dragana.dekanski@inep.ac.rs (D.D.); ninoslavm@inep.co.rs (N.M.); 2Institute of Chemistry, Technology and Metallurgy—National Institute of the Republic of Serbia, University of Belgrade, 11000 Belgrade, Serbia; milena.milosevic@ihtm.bg.ac.rs; 3Department of Pharmacology, Toxicology, and Clinical Pharmacology, Faculty of Medicine, University of Novi Sad, 21000 Novi Sad, Serbia; aleksandar.raskovic@mf.uns.ac.rs (A.R.); nikola.martic@mf.uns.ac.rs (N.M.)

**Keywords:** carob pulp extract, liposomal formulation, encapsulation efficiency, antioxidant activity, controlled release

## Abstract

**Background:** Carob (*Ceratonia siliqua* L.) pulp flour is primarily used in the food industry. As a rich source of bioactive compounds, particularly polyphenols, it holds promise for pharmaceutical formulation research and development. **Objectives:** This study focused on developing liposomal particles loaded with carob pulp extract using the proliposome method, followed by modifications through UV irradiation and sonication. **Methods:** The resulting liposomes were analyzed for encapsulation efficiency, vesicle size, polydispersity index (PDI), mobility, zeta potential, viscosity, surface tension, density, antioxidant activity, FT-IR spectra, and release kinetics under simulated gastrointestinal conditions. In addition, nanoparticle tracking analysis and transmission electron microscopy (TEM) were used for liposomal characterization. **Results:** The findings revealed a high encapsulation efficiency across all samples (>70%). The particle size and PDI measurements confirmed the presence of a multilamellar and uniform liposomal system before post-processing modifications. The medium value of zeta potential suggested a moderately electrostatically stabilized liposomal suspension. The sonicated liposomes demonstrated a higher concentration of vesicles in comparison to non-treated and UV-irradiated samples. TEM analysis revealed purified liposomal vesicles with preserved structural integrity. Encapsulation, as well as UV irradiation and sonication of liposomes, did not diminish the extract’s anti-DPPH activity. However, the ABTS radical scavenging potential of the pure extract was significantly lower compared to its encapsulated counterparts. UV irradiation and sonication notably reduced the anti-ABTS capacity of the extract-liposome system. Monitoring the release of bioactive compounds demonstrated controlled delivery from liposomal particles under simulated gastrointestinal conditions. **Conclusions:** Overall, liposomal formulations of carob pulp extract exhibit significant potential for further development as a functional food ingredient or for use in the prevention and treatment of various diseases.

## 1. Introduction

Carob (*Ceratonia siliqua* L. sub-family Caesalpinioideae, family Leguminosae or Fabaceae) is an evergreen tree, widely cultivated in the Mediterranean region. Its leaves, bark, and seeds have traditionally been used in medicine to treat various diseases, including gastrointestinal disorders, diabetes, and hypertension [1]. The carob powder (flour), obtained by drying, roasting, and grinding the pods, is used primarily in the food industry, in the preparation of sweet juices, chocolates, and biscuits, and as a cocoa substitute [1,2]. It was also found to be rich in phytochemicals and antioxidant compounds, such as polyphenols (phenolic acids, tannins, flavonoids, and their derivatives), reflected in its high antioxidant capacity. Carob pulp has also been recognized as a valuable source of amino acids, minerals, vitamins, sucrose, and insoluble fibers [3]. The plethora of bioactive compounds found in carob fruit and its by-products makes it a promising product and the focus of many studies related to its use in human nutrition and health [4,5]. With their ability to lower triglyceride levels and high antioxidant activity, carob by-products hold great promise as potential therapeutic candidates for the prevention and/or treatment of metabolic syndrome [6]. Specifically, carob pulp has demonstrated anti-diabetic effects, improved glycemic control, enhanced lipid metabolism, and reduced total and LDL cholesterol levels in human studies [7].

Previous research has shown that the extraction method significantly affects the antioxidant capacity of carob pulp extracts [2]. Namely, carob pulp flour extract obtained by microwave-assisted extraction was characterized by a 30% higher yield of total phenolic and total flavonoid content and 30–80% higher antioxidant activity compared with the carob extracts obtained by conventional solid–liquid extraction and ultrasound-assisted extraction. Many phytochemicals with significant therapeutic potential show strong in vitro results but exhibit reduced efficacy in vivo due to their poor ADME (absorption, distribution, excretion, and metabolism) properties [8]. Polyphenols exhibit a bioactivity with a range of 1–10% of ingested doses, highlighting the critical need to enhance their bioavailability [9]. Despite the promising health benefits of dietary polyphenols shown in preclinical studies, the clinical use of polyphenol-based functional foods remains limited due to their low bioaccessibility and/or bioavailability. The encapsulation process protects polyphenols from decomposition throughout the processing and storage stages, prevents their degradation and oxidation in the gastrointestinal environment, and controls their release in the target tissue/organs [10,11]. Liposomal encapsulation of plant bioactives can circumvent the issue of low bioavailability, stability, and water solubility, while the liposome bilayer structure provides the entrapment of hydrophobic, hydrophilic, and amphiphilic agents [12]. Due to the absence of the reaction of phospholipids with taste receptors, liposomes represent appropriate carriers for masking the unpleasant taste of polyphenol components as well [13]. Numerous technologies for liposome preparation were established, such as thin-film hydration, proliposome, detergent removal (depletion), solvent injection, reverse-phase evaporation, lyophilization, supercritical fluid-assisted, supercritical reverse-phase evaporation, supercritical anti-solvent, rapid expansion of supercritical solution, supercritical-assisted liposome formation, microfluidic (channel), membrane contactor methods, etc. [14,15]. Among all the above-mentioned techniques for liposome formulation, the proliposome methodology represents the simplest one, which yields high encapsulation efficiency and is adequate for the preparation of larger quantities of liposomes without employing expensive devices [14,15]. On the contrary, it shows relatively poor reproducibility in preparing a smaller quantity of liposomes [14]. Currently, there is a substantial amount of studies demonstrating the beneficial properties of liposomal encapsulation of plant extracts in terms of their stability and biological potential [11,16,17,18,19]. Thus, the aim of this study was to design different liposomal nutraceutical formulations and evaluate the physicochemical characteristics of carob pulp flour extract-loaded liposomes (non-treated multilamellar vesicles, UV-irradiated, and sonicated). In addition, analysis of the radical scavenging potential of liposomes was performed, and the carob extract polyphenols’ release from the pure extract and encapsulation systems in simulated gastrointestinal conditions was investigated.

## 2. Materials and Methods

### 2.1. Chemicals

Soy L-α-phosphatidylcholine (purity > 95%, phospholipids used for the liposome preparation) was purchased from Avanti Polar Lipids (Alabaster, AL, USA). Ultra-pure water (used for the preparation of the liposomes and their dilution) was from the Simplicity UV^®^ water purification system (Merck Millipore, Darmstadt, Germany). Carob pulp flour (used for the extract preparation), commercially available in Serbia, was purchased from Aroma začini, D.O.O. (Futog, Serbia), while the product was cultivated in Croatia. The following chemicals were also employed: ethanol (Fisher Scientific, Loughborough, Leicestershire, UK), 2,2′-azino-bis(3-ethylbenzothiazoline-6-sulphonic acid)—ABTS, 2,2-diphenyl−1-picrylhydrazyl—DPPH, paraformaldehyde, and glutaraldehyde (Sigma-Aldrich, Hamburg, Germany), and potassium persulfate (used for the activation of ABTS radicals) (Centrohem, Stara Pazova, Serbia). Hydrochloric acid, sodium hydroxide, sodium chloride, and monopotassium phosphate used for in vitro gastrointestinal digestion, catalyzed by pepsin (from porcine gastric mucosa), pancreatin—a mixture of amylase, lipase, and protease (from porcine pancreas) and bile salts—a mixture of bile acid sodium salt and cholic acid–deoxycholic acid sodium salt, were from Sigma-Aldrich (St. Louis, MO, USA).

### 2.2. Preparation of Ethanol Carob Extract

Carob extract was obtained using 1 g of carob pulp flour and 40% ethanol (10 mL) in the microwave-assisted extraction (power of microwaves was 800 W) for 25 min, as described in the previous study [20]. The prepared carob extract was evaporated to dryness, dissolved in water, and stored in a refrigerator until the preparation of liposomal particles. The chemical/phenolic profile of the optimized carob extract, determined using high-performance liquid chromatography with diode-array detection (HPLC-DAD), was also presented previously [2,20]. Briefly, detection was performed at three wavelengths to cover different phenolic groups: 280 nm (for gallic acid, caffeic acid, and *trans*-cinnamic acid), 330 nm (for *p*-coumaric acid, chlorogenic acid, rosmarinic acid, ferulic acid, and quercetin), and 350 nm (for rutin and quercitrin). The analytical quantification was carried out using external standards of the phenolic compounds and was based on peak areas from corresponding standards run under identical chromatographic conditions. All measurements were conducted in triplicate and reported as mean ± standard deviation. A total of seven phenolic compounds were successfully identified and quantified in the sample. Gallic acid was the most abundant (0.46424 ± 0.06964 mg/g dry extract), followed by caffeic acid (0.05353 ± 0.00268 mg/g dry extract), quercitrin (0.03967 ± 0.00198 mg/g dry extract), *p*-coumaric acid (0.02386 ± 0.00239 mg/g dry extract), rutin (0.01883 ± 0.00151 mg/g dry extract), chlorogenic acid (0.01667 ± 0.00083 mg/g dry extract), and quercetin (0.00318 ± 0.00022 mg/g dry extract). *Trans*-cinnamic acid, rosmarinic acid, and ferulic acid were below the limit of detection, which were 0.00003, 0.00015, and 0.00009 mg/g of dry extract, respectively [20].

### 2.3. Development and Modification of Carob Extract-Loaded Liposomes

Carob extract-loaded liposomes were obtained using the previously published proliposome technique, and the established ratio between phospholipids and ultrapure water in our previous study that dealt with rosehip extract-loaded liposomes [21]. In addition, the amount of added carob extract in a liposomal formulation was determined in a preliminary screening, and the selection was made based on the highest encapsulation efficiency. Namely, carob extract in the amount of 40 mL was mixed with 4 g of phospholipids and 12 mL of ethanol. The mixture was heated to 60 °C for 45 min until a homogenous mixture was formed, and ultrapure water was added (80 mL in small portions) during the stirring at 800 rpm for 2 h at room temperature to form multilamellar vesicles. Subsequently, the obtained vesicles were exposed to two modification treatments (UV irradiation and sonication). To examine the effect of UV irradiation treatment on the physicochemical characteristics, antioxidant capacity, and release kinetics of prepared liposomes with carob extract, a liposomal sample in the amount of 20 mL was exposed to UV-C irradiation (253.7 nm, AC2-4G8, ESCo, Singapore). The sample was irradiated in uncovered Petri dishes for 30 min. The liposomes (20 mL) were also sonicated using an ultrasound probe Sonopuls (Bandelin, Berlin, Germany) at 70% amplitude, for 30 min employing a program of 40 s on and 10 s off. The sample was cooled using the ice coating of the flask with the sample during ultrasound treatment. The principal scheme of the development and modification of carob extract-loaded liposomes is presented in Figure 1. All prepared samples were stored at 4 °C until future experiments.

### 2.4. Encapsulation Efficiency Measurement

The values of the encapsulation efficiency of all developed liposomes with carob extract (non-treated, UV-irradiated, and sonicated samples) were determined using the spectrophotometric method. In specific, prepared liposomes were centrifugated at 17,500 rpm and 4 °C for 45 min in Thermo Scientific Sorval WX Ultra series ultracentrifuge (Thermo Fisher Scientific, Waltham, MA, USA) for non-treated and UV-irradiated samples, and ultracentrifuged at 10,000 rpm and 4 °C for 2 h (Optima L-90K Ultracentrifuge, Beckman Coulter, Brea, CA, USA) for sonicated parallel. The concentration of carob extract polyphenols in the supernatants was measured spectrophotometrically at 270 nm (UV Spectrophotometer UV-1800, Shimadzu, Kyoto, Japan).

The encapsulation efficiency was calculated by the amount of carob extract polyphenols in the supernatant, as shown in Equation (1):(1)$$Encapsulation\;efficicency\left(\%\right)=\frac{C_{i}-C_{sup}}{C_{i}}\times 100$$
where C_i_ is the initial content of carob extract polyphenols used for the preparation of liposomes, and C_sup_ is the content of carob extract polyphenols determined in the supernatant.

### 2.5. Analysis of Rheological Properties of the Liposomes

The rotation viscometer Rotavisc lo-vi (IKA, Staufen, Germany), frequency of 50/60 Hz, was employed for the measurement of liposome viscosity. Namely, non-treated, UV-irradiated, or sonicated liposomes in a volume of 6.7 mL were placed in a VOL-C-RTD chamber with VOLS-1 adapter and cylinder stainless steel spindle (single VOL-SP-6.7 spindle), and the measurements were performed in three repetitions at 25 °C on the 1st and 60th days. The viscosity was measured with constant rotation at 200 rpm, and the deflection, as a measurement of the torque, was >30 M%.

The surface tension and density of all three types of developed liposomes in a volume of 20 mL were measured using the device Force Tensiometer K20 from KRÜSS (Hamburg, Germany). The surface tension was determined using the Wilhelmy plate (range from 1 to 999 mN/m and resolution of 0.1 mN/m), while density was determined using a silicon crystal as the immersion body. The measurements were performed in three repetitions for density and at least three repetitions for surface tension at 25 °C on the 1st and 60th days, as in the case of viscosity determination.

### 2.6. Storage Stability Study

The size, polydispersity index (PDI), mobility, and zeta potential of all developed carob extract-loaded liposomes (non-treated, UV-irradiated, and sonicated) were determined using dynamic light scattering (DLS) in the Zetasizer Nano Series device (Malvern Instruments, Malvern, UK). The measurements were performed on the 1st, 7th, 21st, 28th, and 60th days after the preparation and modification of the liposomes during the 60-day storage in a refrigerator (4 °C). For the DLS analyses, the liposomal emulsion was diluted 500 times, and the measurements were performed in three repetitions at 25 °C.

### 2.7. Nanoparticle Tracking Analysis

The concentration and size distribution of liposome preparations were assessed using the nanoparticle tracking analyzer (NTA) ZetaView Quatt PMX-430 equipped with ZetaView software version 8.05.16 SP3 (Particle Metrix, Inning am Ammersee, Germany). Prior to analysis, the instrument underwent an automatic cell check followed by alignment of the camera and laser. Focus was confirmed using 100 nm polystyrene beads according to the manufacturer’s guidelines. Liposome preparations were diluted in deionized water to achieve optimal particle count per frame. Between each measurement, a washing step was performed. Measurements in light scatter mode (LSM) were conducted by exposing liposomes to a blue laser (488 nm). Video acquisition settings included a shutter speed of 100 and a frame rate of 30 per cycle, with sensitivity set to 78. Post-video-capturing parameters were adjusted to a minimal area: 10; maximal area: 1000; minimum brightness: 30. Samples were analyzed three times at up to 11 positions.

### 2.8. Fourier Transform Infrared (FT-IR) Spectroscopy

FT-IR spectroscopy was employed to record spectra of pure phospholipids (used for the liposome preparation), pure carob extract, and three types of developed liposomes (non-treated, UV-irradiated, or sonicated parallels) with the aim of investigating potential chemical changes during liposome formation and modification, or incompatibility between phospholipids and compounds from carob extract, as well as the effectiveness of the encapsulation, i.e., entrapment of carob bioactives by liposomal vesicles. Since the device required the samples without water, pure extract and liposomal formulations were freeze-dried in the lyophilization equipment Alpha 2–4 LSCplus (Christ, Osterode am Harz, Germany). Specifically, centrifuged liposomes (with excluded non-encapsulated fraction of carob extract) were frozen and freeze-dried at 0.011 mbar and −75 °C for 24 h. The FT-IR spectroscopy was performed in the wavenumber range between 400 and 4000 cm^−1^, employing the device Nicolet™iS™10spectrometer from Thermo Fisher Scientific (Waltham, MA, USA) equipment with Smart iTR™ Attenuated Total Reflectance (20 scans mode, resolution of 4 cm^−1^).

### 2.9. Transmission Electron Microscopy (TEM)

Liposome preparations (10 µL each) were applied to formvar-coated copper grids, 200-mesh, for 30 min, at room temperature. After that, fixation with 2% paraformaldehyde was performed for 10 min by grid flotation, followed by washing three times with deionized water. Post-fixation was performed with 2.5% glutaraldehyde for 5 min, followed by rinsing with deionized water for 5 min. The grids were dried at room temperature, after which images were captured using a Philips CM12 electron microscope (Philips, Eindhoven, The Netherlands).

### 2.10. Analysis of the Radical Scavenging Potential of Liposomes (DPPH and ABTS Assays)

The radical scavenging potential of the capacity of non-treated, UV-irradiated, and sonicated carob extract-loaded liposomes and pure extract (diluted to achieve the same concentration as in liposomes) was investigated employing the DPPH and ABTS radical scavenging assays.

In the DPPH assay [22], ethanol DPPH solution (the absorbance of ~0.800) in a volume of 2 mL was added to the 20 µL of liposomal emulsion or diluted carob extract. The absorbance was read at 517 nm after incubation at room temperature in the dark for 20 min. The antioxidant capacity of the samples was calculated using the following Equation (2):(2)$$Radical\;scavenging\;capacity(\%)=(A_{c}-A_{x})\times 100/A_{c}$$
where A_c_ is the absorbance of the control (DPPH solution and water) and A_x_ is the absorbance of the DPPH solution and liposomes or extract. The analysis was performed in three replicates, and the antioxidant potential was shown as the percentage of DPPH scavenging.

In the ABTS test [23], ethanol ABTS solution (the absorbance of ~0.700) in a volume of 2 mL was added to the 20 µL of liposomal emulsion or diluted carob extract. The absorbance was read at 734 nm, after the incubation at room temperature in the dark for 6 min. The antioxidant capacity of the samples was calculated using the following Equation (3):(3)$$Radical\;scavenging\;capacity(\%)=(A_{c}-A_{x})\times 100/A_{c}$$
where A_c_ is the absorbance of the control (ABTS solution and water) and A_x_ is the absorbance of the ABTS solution and liposomes or extract. The analysis was performed in three replicates, and the antioxidant potential was shown as the percentage of ABTS scavenging.

### 2.11. In Vitro Release Kinetics (Franz Diffusion Cell)

The in vitro release kinetics from pure carob extract and three types of developed liposomes with carob extract (non-treated, UV-irradiated, and sonicated samples) were monitored in a Franz diffusion cell donated by PermeGear, Inc. (Hellertown, PA, USA). Carob polyphenol release from extract and liposomes under simulated gastrointestinal conditions was investigated at 37 °C using an acetate cellulose membrane for the separation of two compartments (donor and acceptor cells) and simulated gastric fluid (SGF) or simulated intestinal fluid (SIF). In vitro gastrointestinal digestion catalyzed by pepsin and pancreatin was performed employing the protocol described by Liu et al. [24] with some modifications. SGF was obtained by dissolving NaCl (0.4 g) in 6 M HCl solution (1 mL) and adding distilled water (160 mL). The pH value was adjusted to 1.5, and the volume to 200 mL. The stock solution was incubated at 37 °C (30 min), and pepsin was dissolved in the pre-heated stock solution (3.2 mg/mL) with constant stirring. SIF was obtained by dissolving K_2_HPO_4_ (3.4 g) in 0.1 M NaOH solution (95 mL) and adjusting the pH to 7.4 and the volume to 500 mL. Subsequently, the bile salts were added (0.2 mg/mL), and the solution was incubated at 37 °C (30 min) with constant stirring. Pancreatin was dissolved in the pre-heated stock solution (3.2 mg/mL). The sample (pure extract or liposomes, 2 mL) was placed in the donor cell on the acetate cellulose membrane, while the receptor cell was filled using SGF or SIF. The medium was mixed at 850 rpm by magnetic stirring at 37 °C using a water jacket and a peristaltic pump. The polyphenol release from the extract or liposomes was monitored for 3 h (SGF) and 8 h (SIF). The samples (500 µL) were taken from the receptor cell at certain time intervals, and the polyphenol concentration was determined spectrophotometrically at 270 nm [25]. The diffusion of polyphenols to the receptor fluid through the membrane was approximated using Fick’s second law, and the calculation flow related to the diffusion of polyphenols from liposomes and extract, diffusion coefficients, and diffusion resistance with corresponding formulas and explanations, is presented in the Appendix A.

### 2.12. Statistical Data Processing

In order to determine the presence of statistically significant differences between the samples, one-way analysis of variance and Duncan’s post hoc test (STATISTICA 7.0) were employed for statistical data processing. The results in the tables and graphs are presented as mean ± standard deviation, and all analyses were performed in triplicate (n = 3), while the differences among samples were considered significant at *p* < 0.05. The data obtained in the analyses of encapsulation efficiency, size, PDI, mobility, zeta potential, rheological properties, and anti-radical potential of developed liposomes were subjected to statistical processing using the above-mentioned tools.

## 3. Results and Discussion

The present research paper provides evidence related to physicochemical properties, stability, antioxidant capacity, and release kinetics of developed carob extract-loaded liposomes, as well as the influence of UV irradiation and sonication on the mentioned variables.

### 3.1. Encapsulation Efficiency of Developed Liposomal Vesicles

With the aim of investigating the success of the encapsulation process of active principles from carob extract in developed liposomal formulations, liposome encapsulation efficiency was quantified, and the data are presented in Table 1. The mass of encapsulated carob polyphenols was determined indirectly by measuring the concentration of non-encapsulated polyphenols in the supernatant of developed liposome suspensions.

Carob polyphenols were successfully encapsulated in liposomes employing the proliposome technique, with an encapsulation efficiency of 80.59 ± 1.29% (Table 1). The data shown in Table 1 also indicate that post-preparation modifications significantly changed (decreased) encapsulation efficiency. Namely, the encapsulation efficiency was 74.99 ± 1.02% in the UV-irradiated formulation and 71.05 ± 1.34% in the sonicated formulation (Table 1). Various studies on the encapsulation of plant extracts in liposomes report encapsulation efficiency in a wide range, from ~40% to ~95%, depending on liposome composition and encapsulated plant-origin compounds [11,18,19,26,27]. For example, in our previous study, the entrapment efficiency of ethanol rosehip extract in multilamellar liposomes was 90.8% [21], while similar values were found for the encapsulation of turmeric extract (over 85%) [27]. However, the encapsulation efficiency of carob polyphenols in the present study was lower and similar to the values obtained for betel ethanolic extract (71.8–79.9%) [28], palm seed extract (71.0–86.8%) [19], pineapple mint extract (73.3–77.5%), and pokeweed extract (81.4–84.0%) encapsulated in liposomes [29]. The obtained differences are expected due to the proven significant impact of phospholipid and/or sterol contents on encapsulation efficiency [30]. According to the data in the literature, increasing the phospholipid concentration enhances the incorporation capacity for plant extract within liposomal vesicles [26]. Hence, several studies have reported that in the case of decreased concentration of phosphatidylcholine, the liposomal particles available for the encapsulation of phenolic compounds were limited. For example, in the studies of Takahashi et al. [27] and Lu et al. [17], the encapsulation efficiency of plant extracts increased with the increase in lecithin concentration. Since the concentration of phospholipids in developed liposomes with carob extract was high (5%), it can explain the high values of the entrapment efficiency for carob polyphenols.

Furthermore, the stability of liposome particles can be reduced or degraded because of lipid or other compound breakdown or oxidation caused by exposure to UV light. It can alter the liposomal bilayer structure by affecting the interactions among hydrophobic and hydrophilic regions [31]. The mentioned alterations can trigger modifications in the liposome’s stability, permeability, and other characteristics as well. Therefore, the changes in the encapsulation efficiency value after UV treatment were also expected. Namely, the decrease in the encapsulation efficiency of carob polyphenols after irradiation by UV light can be explained by the leakage of encapsulated compounds upon irradiation. UV irradiation is known to generate reactive oxygen species (ROS) that can make pores on the phospholipid bilayer, causing UV-induced liposome damage, i.e., a significant leakage rate, which was already shown in the literature [32]. FT-IR analysis of liposomes with carob extract (shown in Section 3.4) also confirms the occurrence of oxidation/peroxidation processes after UV light exposure. Additionally, the destabilization of the phospholipid bilayer and, consequently, release of the entrapped bioactives can occur due to the polymerizable domain contraction or polymerized domain distortion because UV lights directly initiate the phospholipid polymerization [32]. After irradiation, covalent bonds are created in the hydrophobic tails, binding the polymerizable lipids together. Also, the polymerized domain cannot be incorporated into the phospholipid bilayer membrane, forming micelles or disks with unfavorable interactions among the hydrophilic head groups (from the poly (lipid) domains) and the lipophilic tails (from the phospholipid bilayer membrane). Thus, upon UV light treatment, diffusion of the poly (lipid) domain induces the leakage of the encapsulated active principles [32].

Lower encapsulation efficiency after the sonication of carob extract-loaded liposomes was most likely influenced by the breakdown of liposome membranes under the sonication method via the cavitation mechanism. Further, polyphenols entrapped between the phospholipid bilayers may be partially released, which could result in decreased encapsulation efficiency [33]. Chotphruethipong et al. study [33] showed that higher levels of amplitude contributed to the higher reduction in liposome encapsulation efficiency at all employed periods of the sonication process. In the present study, a 70% amplitude of the ultrasound probe was employed. Machado et al. [34] have also reported that the liposomal vesicles subjected to homogenization provided 10% higher encapsulation efficiency in comparison to those subjected to ultrasound waves, as in the case of liposomes with carob extract. The mentioned results can be due to the excessive amount of energy delivered to the liposomal system, disrupting the liposomal particles [33,35]. Additionally, the temperature in the system increased during the sonication process (despite the ice coating of the flask with the liposome sample), consequently increasing the liposome membrane fluidity and inducing a higher polyphenol release from the liposomal vesicles, thus lowering the encapsulation efficiency [35]. The two key limiting factors for the utilization of ultrasound waves in the preparation of liposomes are low and restricted encapsulation efficiency, as well as the lack of reproducibility throughout various systems.

After 60 days of storage, encapsulation efficiency did not change in non-treated liposomes (78.84 ± 1.95%, Table 1), while a significant drop was noticed in UV-treated and particularly in sonicated liposomes (69.51 ± 0.91% and 48.62 ± 1.78%, respectively). The obtained results were expected due to the above-mentioned changes that occur during post-processing treatments, resulting in a more fluid membrane, as well as facilitated leakage of carob polyphenols from liposomes.

### 3.2. Size, Size Distribution, Mobility, and Zeta Potential of Developed Liposomal Vesicles

As can be seen from Figure 2A, on the 1st day, non-treated liposomes with carob extract had a diameter of 2055 ± 120 nm, while UV irradiation treatment caused a significant increase in the mentioned parameter (3662 ± 131 nm). As expected, the sonication treatment significantly reduced the size of developed liposomes with carob extract up to 194.0 ± 11.0 nm (Figure 2A). A graphical presentation of row data obtained using the dynamic light scattering method is shown in Appendix A. On the 1st day, the PDI value for the non-treated sample was low (0.137 ± 0.021), whereas both modification treatments resulted in an increase in the PDI values of the liposomal system, 0.329 ± 0.024 (for UV-irradiated) and 0.459 ± 0.068 (for sonicated) (Figure 2B). The mobility was −2.26 ± 0.02 µmcm/Vs for non-treated, −1.98 ± 0.07 µmcm/Vs for UV-irradiated, and −1.06 ± 0.12 µmcm/Vs for sonicated liposomes on the 1st day (Figure 2C). The zeta potential possessed negative values for all developed liposomal systems: −28.6 ± 0.3 mV for non-treated, −25.2 ± 0.9 mV for UV-irradiated, and −13.9 ± 1.5 mV for sonicated liposomes (measured immediately after the preparation, Figure 2D). Both UV light and ultrasound waves cause a significant drop in the mobility and zeta potential of liposomes with carob extract.

The development of large liposomes with carob extract can be explained by the presence of carob polyphenols. Namely, polyphenols have a negative charge; therefore, higher extract content results in increased electrical repulsive forces and liposome size, consequently causing structural instability of developed lipid particles [11]. In the case of higher encapsulation efficiency (which was also in carob extract-loaded liposomes), with a decreasing volume of liposome core, the number of polyphenol compounds located interior liposomal bilayer was high, and accordingly, the diameter of the liposomal particles increased [11,17]. The study of Jahanfar et al. [11] also showed that the placement of non-polar compounds in the liposomal bilayer can affect the increase in vesicle diameter, as well as a rupture in their membrane, which can also result in enhanced permeability. The obtained value of liposome diameter was in agreement with our previous study, where rosehip oil-loaded liposome size was also high, 2145.7 nm [36]. However, the size of natural extracts-loaded liposomes after sonication at 70% amplitude for 30 min (as in the case of sonicated liposomes with carob extract) was significantly lower (61.4 nm) compared to our result (194 nm). The reason can lie in differences in lipid compounds and their concentration, the nature of the extracts, and the initial vesicle size. Nevertheless, sonicated grape pomace extract-loaded liposomes had a size of 245 ± 26 nm [37], while the ginger ethanolic extract loaded-nanoliposome size was 164.5 nm without sonication [16]. The increased liposome size under UV irradiation was also proven in the literature, suggesting the liposome photochemical destruction across the photon energy absorption, which results in drastic alterations of the bilayer conformation [38]. UV light exposure triggers the changes in the physical characteristics of phospholipid bilayers via disturbing phospholipid order and packing, thus inducing a size increase, as well as modifications of membrane permeability [38]. Additionally, as a result of the dominant effect of UV irradiation on the liposome membrane damage, i.e., photodegradation, highly disordered chains of polyunsaturated fatty acids and consequently exposed hydrophobic patches can promote vesicle aggregation as well [38]. According to Freitas and Müller [39], the introduction of energy, including temperature or light, to the liposome system led to vesicle growth accompanied by a lowering in zeta potential values (which is shown for carob extract-loaded liposomes as well) and gelation.

The measured value of size distribution agrees with the study of Ganji et al. [16], where the PDI of liposomes with ginger extract amounted to 0.186. PDI values equal to or less than 0.2 are acceptable for polymer-based carriers, while in applications of liposomes for drug delivery, PDI values of 0.3 and below can be acceptable. Additionally, post-formation processing of liposomes with carob extract (UV irradiation and sonication) significantly affected the mentioned variable, causing its increase. The measured values agree with the literature data, where the PDI values of sonicated liposomes with encapsulated doum fruit extract were between 0.425 and 0.469 [40], and for sonicated rosehip oil-loaded liposomes, PDI was 0.439 [36], as in the case of sonicated liposomes with carob extract. The highest PDI was recorded for the sonicated sample, which indicates the presence of multilamellar vesicles along with small unilamellar vesicles. It was also shown in a graphical presentation of row data obtained using the DLS method for sonicated carob extract-loaded liposomes (Appendix A), where apart from small unilamellar vesicles (size of around 100 nm), peaks after 1000 nm appear as well, causing an increase in the PDI values. Moreover, larger liposomes (with vesicle sizes exceeding 1000 nm) tended to exhibit lower PDI values compared to smaller liposome populations (ranging from 100 to 400 nm in size) [41]. Zeta potential serves as a crucial parameter for assessing the colloidal stability of liposomal dispersions, as it reflects the extent of electrostatic repulsion between nanoparticles within the system. Particles with high absolute zeta potential values, whether positive or negative, are well stabilized due to strong repulsive forces, whereas those with low zeta potentials are more prone to aggregation or flocculation [42]. Since the absolute values of the mentioned variables were the lowest for sonicated samples (shown later in Figure 2D), this can explain the formation of aggregates, apart from smaller vesicles, which resulted in a higher size distribution, i.e., PDI values. Lower mobility values for sonicated vesicles (shown later in Figure 2C) also confirm this claim of the aggregate occurrence, since aggregates show reduced mobility. UV light exposure also significantly increased the size heterogeneity of the liposomal system with carob extract, which can be explained by the above-mentioned modifications of the membrane conformation and liposome size, as well as the potential aggregation of lipid particles [38].

The liposome mobility measurement is important to evaluate their stability in an aqueous medium. Namely, high mobility values lead to a repulsion among liposomal vesicles, preventing vesicle aggregation and proving liposome system stability [43]. Additionally, mobility value is important to formulate adequate drug delivery systems (including liposomes) under optimized encapsulation conditions, figure out mechanisms of drug release, and predict their behavior in vivo [44]. According to the literature data, the values of liposome mobility varied in a wide range depending on the used phospholipids, encapsulated compounds, and liposome size [12,43]. The data of carob extract-loaded liposome mobility shows a negative value of mobility because phospholipids are negatively charged. Jacquot et al. study [43] demonstrated that higher mobility possessed larger liposomes, which was also the case with non-treated and UV-treated liposomes with carob extract (larger liposomes with higher mobility in comparison to smaller sonicated parallels). Furthermore, UV-irradiated and sonicated liposomes possessed a significantly lower value of encapsulation efficiency; therefore, the non-encapsulated fraction of carob polyphenols can be located on the outer surface of liposomes due to proven interactions between lipids and polyphenols [45], which consequently can reduce the liposome mobility.

Zeta potential, as one of the main parameters in the investigation of the stability of liposomal formulations, should be higher than +30 mV or lower than −30 mV with the aim of preventing or minimizing liposome aggregation and fusion of liposomal particles [11,17]. Hence, higher values of zeta potential of the liposomal system demonstrate its higher stability due to more repulsive forces that prevent vesicle aggregation and fusion. However, the measured zeta potential of all developed liposomes with carob extract was at a middle level. A wide range of different zeta potential values obtained in numerous studies depends on various characteristics of phospholipid molecules used in liposome structures [11]. In addition, circumferential temperature and ionic power affect the positive and negative values of zeta potential. In the case of lower circumference ionic power, phosphatidyl groups are located in the outer part of the polar head of phospholipids, resulting in a negative zeta potential of vesicles [11,17], as in our study. As outlined by Ben-Fadhel et al. [46], sonication treatment significantly decreased the zeta potential of natural extracts-loaded liposomes by 6–10 mV, as in the case of carob extract-loaded liposomes. Freitas and Müller’s study [39] reported that the consequence of changing the liposomal system after UV irradiation is the reduction in the values of zeta potential, which can also be the reason for a higher possibility of aggregation occurrence. Namely, supported by a decreased zeta potential, i.e., sufficient reduction in repulsive forces, lipid vesicles may interact and form a gel network [38].

Figure 2A also shows that in the first 14 days of storage, the diameter of all developed liposomes did not change (2187 ± 96 nm for non-treated, 3805 ± 147 nm for UV-treated, and 236 ± 35 nm for sonicated liposomes). However, up to the 60th day, the size of liposomal particles increased and amounted to 4085 ± 139 nm for non-treated, 4902 ± 114 nm for UV-irradiated, and 354 ± 62.5 nm for sonicated liposomes. PDI values did not alter during a two-week period as well: 0.186 ± 0.062 for non-treated, 0.351 ± 0.054 for UV-irradiated, and 0.469 ± 0.045 for sonicated sample, while changes were noticed on the 21st day of the measurement: 0.325 ± 0.032, 0.510 ± 0.061 for UV-irradiated, and 0.746 ± 0.043, respectively (Figure 2B). In higher phospholipid concentrations (as in carob extract-loaded liposomes), more phospholipid molecules entered into each liposomal particle; thus, the diameter and instability of the liposome structure increased [11,30]. Namely, the employment of high phospholipid content reduced their solubility and enhanced the possibility of phospholipid aggregation because of the more consumed process energy necessary for the distribution of phospholipids compared to smaller vesicles [30]. Additionally, the drop in the viscosity values of developed carob extract-loaded liposomes after storage (shown in Table 1) can be responsible for liposome system instability: alteration/increase in the particle size, i.e., potential aggregation of lipid vesicles, and consequently creating a more heterogeneous system, i.e., enhancement in the values of polydispersity [47,48].

Mobility and zeta potential values did not modify in the first 7 days of storage (Figure 2C and Figure 2D, respectively) but lower values were measured after 14 days of storage at 4 °C (−2.03 ± 0.04 µmcm/Vs and −25.8 ± 0.5 mV for non-treated, −1.80 ± 0.10 µmcm/Vs and −23.8 ± 1.56 mV for UV-irradiated, as well as −0.80 ± 0.10 µmcm/Vs and −10.3 ± 0.8 mV for sonicated particles). After that period, no additional changes in the zeta potential values of all developed liposomes were noticed on the 60th day (−25.7 ± 1.0 mV for non-treated, −24.7 ± 1.4 mV for UV-irradiated, and −11.2 ± 0.5 mV for sonicated particles). However, during the 60-day storage study, zeta potential values of post-processing modified liposomes varied in a narrow range in comparison to the non-treated sample. The same trend was observed for the mobility after 60 days: −2.11 ± 0.08 µmcm/Vs for non-treated, −2.00 ± 0.09 µmcm/Vs for UV-treated, and −0.91 ± 0.17 mV for sonicated systems. Wolfram et al.’s study [12] also showed a slight alteration in the mobility and zeta potential of phospholipid liposomes after two weeks. The fact that the noticed alterations in mobility and zeta potential of liposomes with carob extract (after 7 days of storage) had no effect on the liposome size (without significant changes during 14 days) can be explained by the more significant contribution of zeta potential to the liposomal mobility than to the liposome size.

### 3.3. NTA Measurements

NTA was used to determine the size distribution and concentration of non-treated, UV-irradiated, and sonicated carob extract-loaded liposomes, and the data are presented in Figure 3.

NTA of liposome size distributions indicated that the non-treated and UV-irradiated preparations were more heterogeneous in size (30–600 nm) compared to the sonicated liposomes (30–300 nm), with the non-treated sample exhibiting the greatest size disparity. The sonicated liposomes demonstrated the highest concentration of vesicles per milliliter (1.30 × 10^13^) compared to the other two preparations (over 10-fold greater, 1.03 × 10^12^ for the non-treated sample and 8.17 × 10^11^ for the UV-irradiated sample).

### 3.4. Rheological Properties of Developed Liposomal Formulations

The viscosity of the liposomal system has an essential role in its long-term storage and represents a relevant factor of stability and controlled release of encapsulated components [49]. Therefore, the viscosity of developed liposomal formulations with carob extract was measured on the 1st and 60th days of storage at 4 °C. Initial data on viscometry measurements are presented in Appendix A. The results are presented as mean ± standard deviation in Table 1. The values of viscosity of non-treated and UV-irradiated liposomes with carob extract amount to 18.40 ± 1.22 mPa·s and 16.90 ± 0.85 mPa·s, respectively, while the sonicated sample showed significantly lower viscosity (5.17 ± 0.09 mPa·s). The study of Nareni et al. [48] showed that liposomes of smaller size possessed a lower viscosity, as in the case of sonicated carob extract-loaded liposomes. Karaz and Senses [49] also reported that the liposomes with a high fraction of multilamellar structures had increased viscosity, while the viscosity was lower for the unilamellar vesicles due to the reduction in volume fraction. According to the literature data, the increased particle size of liposome formulations can result in increased viscosity and, hence, higher PDI values. Namely, an increase in liposome size can induce the formation of complex linkages and crystallinity of particles in the central core because of their solid property, contributing to higher viscosity values of the liposome system. Higher concentrations of phospholipids in liposomal formulation, such as lipid-based vesicles (which is also the case with developed carob extract-loaded liposomes), can lead to higher aggregation because of decreased steric stabilization and increased molecular interactions, consequently resulting in a higher viscosity value [47]. Changes in the liposome viscosity can also induce changes in liposome stability, potential separation or sedimentation, Brownian motion of particles in the system, delivery of bioactives from liposomes, etc. [48]. As can be seen from the results of the stability study (Figure 2), in all liposome samples (with lower and higher viscosity values), alterations in physical properties, such as size and size distribution, have occurred during storage. Hence, during the 60-day storage at 4 °C, the viscosity of developed liposomes significantly decreased: 13.93 ± 0.82 mPa·s for non-treated, 14.20 ± 1.03 mPa·s for UV-irradiated, and 4.46 ± 0.14 mPa·s for sonicated samples (Table 1). The decrease in viscosity of all developed liposomal systems can be attributed to the occurrence of the reactions of hydrolysis that are specific to the water environment, also presented in carob extract-loaded liposomes [36].

Surface tension has an important role in interfacial dynamics, water penetration within the phospholipid membrane bilayers, as well as lateral phospholipid packing of liposomes [50]. Therefore, the surface tension of developed liposomal preparations with carob extract was determined on the 1st and 60th days of storage at 4 °C (Table 1). Initial data on tensiometry measurements are presented in Appendix A. The surface tension of non-treated and UV-irradiated liposomes was 41.75 ± 1.98 mN/m and 39.57 ± 1.06 mN/m, respectively, while the sonicated liposomes possessed a significantly lower surface tension value, 24.83 ± 1.27 mN/m. The surface tension of all prepared liposomes altered during 60 days, showing a decreasing trend. Namely, the measured values were significantly lower in comparison to initial measurements and amounted to 30.20 ± 0.93 mN/m (non-treated), 29.27 ± 1.00 mN/m (UV-treated), and 22.03 ± 0.92 mN/m (sonicated) (Table 1). The presence of various ingredients in water surroundings can trigger structural changes in aqueous mediums. Depending on their influence on the water hydrogen-bonded networks, the mentioned compounds are classified as “structure breakers” and “structure makers” [50]. “Structure breakers” accumulate in the interface and decrease surface tension, while “structure makers” increase the surface free energy, decrease the interfacial area, and consequently increase surface tension compared to the bulk phase. Since the measured surface tension of pure water (the medium used for the liposome preparation) was 69.9 ± 1.16 mN/m, phospholipids belong to the class of “structure breakers” because the surface tension of developed liposomes with carob extract was significantly lower in comparison to water. The obtained phenomenon is in agreement with the literature data, where small unilamellar liposomes reduce the value of water surface tension due to amphiphiles from the liposomal vesicles and spreading as a monomolecular layer at the interface between the high-dielectric-constant polar water and the low-dielectric-constant nonpolar oil [51]. At higher surface tension levels, there is a reduced number of interfacial water molecules because of the increased lipid packing [50]. Thus, it can be concluded that non-sonicated liposomes with carob extract (multilamellar liposomal vesicles) possessed augmented lipid packing compared to reduced-size, sonicated liposomes. However, the mentioned decrease in the mean molecular area per lipid (augmented packing) can be reduced with the increase in the repulsion at the headgroups and acyl chains [50]. Rivera et al. demonstrated that lipid packing and surface tension affect the zeta potential of liposomes as well [52]. In the case of liposomes with encapsulated carob polyphenols, the sample with a lower zeta potential also possessed a lower surface tension (sonicated system). The decrease in the surface tension value of sonicated liposomes with carob extract should be attributed to impurities (debris) produced by the employed ultrasound probe [53]. TEM analysis of the obtained sonicated liposomal sample with carob extract also showed the presence of the ultrasound probe debris (presented later in Figure 5). Lombardo and Kiselev’s study also reported that a major drawback of the sonication technique is the potential for titanium particle shedding from the probe tip, which can lead to contamination of the lipid formulations [42]. Wei et al.’s study [54] showed the absence of correlation between surface tension and viscosity, as the two fluid characteristics, as well as a significant impact of the intermolecular attraction of water molecules on the surface tension; therefore, the decrease in viscosity did not trigger changes in the surface tension of liposomes. Thus, the alterations in the surface tension of developed liposomes during storage can be explained by the potential occurrence of nanobubbles at the liquid’s surface. Namely, according to the literature, with time, the nanobubble quantity in the bulk liquid significantly reduces, while the number of nanobubbles gradually adsorbed at the liquid surface increases. Therefore, the drop in surface tension values can be attributed to the Janus-like structure of nanobubbles, which can break the water molecules’ hydrogen bonding network at the surface of the liquid [53].

The density of the carob extract-liposome system was also monitored on the 1st and 60th days of storage. As can be seen in Table 1, the density of non-treated and UV-irradiated samples was the same, 1.04 g/cm^3^, whereas the density of the small, ultrasound-treated liposomal population amounted to 1.02 g/cm^3^. However, there was no statistically significant difference between the measured values. The mentioned variable of all developed liposomal populations did not alter from the 1st to the 60th day (Table 1). The absence of a significant difference in the density values of the three liposomal systems was expected since the density of liposomes and other liquid systems is affected by the type and concentration of employed lipids and solvents for their preparation. In the case of carob extract-loaded liposomes, the same type and content of phospholipids, as well as solvent, were used in all samples.

### 3.5. FT-IR Spectroscopic Analysis of Developed Liposomal Formulations

FT-IR spectroscopic analysis was employed to examine potential chemical changes in phospholipids and carob extract compounds that can occur during liposome preparation, as well as changes in lipid bilayers during modifications, such as treatments by UV irradiation or sonication. The data are presented graphically in Figure 4.

The most common route of administration of different compounds loaded in liposomes is per os and parenteral applications; therefore, developed encapsulates must be sterilized employing different processes, such as autoclaving, ultraviolet, or gamma ionizing irradiation, filtration, etc. Some of the mentioned techniques can lead to liposomes’ bilayer destabilization via oxidation of the liposomal membrane, i.e., peroxidation of unsaturated lipids, hydrolysis, lipid fragmentation, and pH changes in the liposome system [31]. Thus, UV-irradiated liposomes were subjected to FT-IR spectroscopy analysis, and their spectra were compared to the non-treated parallels.

The spectrum of the extract displays broad bands at 3295 cm^−1^ assigned to O-H stretching vibrations of hydroxyl, phenolic, and carboxyl groups. The bands in the region 2978–2880 cm^−1^ and bands at 1446, 1418–1375, and 1328 cm^−1^ correspond to stretching vibration of methylene and methyl, bending vibrations of methylene, rocking vibrations of C-H bonds, and bending vibrations of methyl groups, respectively [55]. In addition, a C=C stretching band of aromatic and vinyl parts in the structures of the components of extract appears at 1602 cm^−1^, while asymmetric and symmetric C–O/C-O-C/C-OH stretching vibrations of ether, phenol, alcohol, and alkyl ester groups at 1266–924 cm^−1^, and =CH deformational vibrations of aromatic and vinyl structures at 761–720 cm^−1^ are observed [3]. All these bands confirm the presence of the components of carob extract: flavonoids, phenolic acids, and carbohydrates [1,2].

The FT-IR spectra of L-α-phosphatidylcholine (Ph) show peaks observed at 3012, 2925–2850, and 1464–1333 cm^−1^, related to =CH stretching vibrations, -CH_2_ and -CH_3_ stretching and bending vibration of fatty acid residue [56]. Additionally, the bands observed at 1735 and 1652 cm^−1^ are due to the stretching vibrations of C=O in the ester group and O-H deformations overlapped with C=C stretching vibration in fatty acid residue. The ether C-O-C/C-O and phosphate P=O/P-O-C groups were observed in the 1261–965 cm^−1^ region [56]. The peaks at 874 and 720–500 cm^−1^ are related to P-O asymmetric stretching vibration and =C-H out-of-plane in *cis*-unsaturated double bond.

The spectra of liposomes, i.e., non-treated, UV-irradiated, and sonicated samples, indicated the presence of characteristic peaks from phospholipids and extract. In addition, intra/intermolecular interaction in the structure of phospholipids and intermolecular/hydrogen bonding interaction between phospholipids and extract lead to a change in the structure of the peak (intensity and shape) or peak shifting. The spectra of non-treated and sonicated liposomes are similar, while the spectrum of UV-irradiated liposomes shows spectral change, as observed in Figure 4. The addition of extract to the phospholipids leads to a decrease in bands assigned to hydroxyl, carboxyl, ethylene, methylene, methyl, and ether groups for non-treated and sonicated liposomes. In addition, new bands, observed at 1609, 1133, and 924 cm^−1^, appear as a result of the extract addition. The peak shifting from 1261 cm^−1^ in phospholipids to 1236 cm^−1^ in liposomes and the disappearance of the bands at 1652, 1071, and 1091 cm^−1^ indicate the incorporation of the extract into the phospholipid liposome structure and the creation of different intermolecular interactions. The opposite trend of the peak structure change in the region 1200–965 cm^−1^ is observed for liposomes treated by UV irradiation. In this region, the intensity of the peak at 1045 cm^−1^ increases, and a new band at 990 cm^−1^ appears. Also, the increase in broadband for aromatic and vinyl structures, observed at 1609 cm^−1^, is noticed. This increase in intensity peaks is probably related to oxidation/peroxidation processes, causing the formation of oxygen-containing functionalities. Further, small peaks at 874 and 816 cm^−1^ suggest the presence of hydroperoxide species. Accordingly, UV irradiation and ultrasound treatment of liposomes caused a decrease in the double bond at 3012 m^−1^, a change in the structure of peaks between 1735 and 1600 cm^−1^, and the existence of small bands at 874–816 cm^−1^ as a result of forming low-stability oxygen species. The results obtained from FT-IR analysis are consistent with radical scavenging activity analysis (Section 3.5), where non-treated liposomes with extract showed higher antioxidant capacity in the ABTS assay compared to post-processing modified liposomes.

### 3.6. TEM Analysis

TEM analysis of non-treated, UV-irradiated, and sonicated liposome preparations loaded with carob extract revealed purified liposomal vesicles with preserved structural integrity (Figure 5). All three preparations exhibited size heterogeneity, mostly ranging from 80 to 500 nm, with the non-treated sample displaying the most pronounced variability. A small number of liposomes in each preparation exceeded 1–2 µm in diameter. According to the TEM images, the irradiated vesicles showed a comparable size to the sonicated ones. Also, the TEM analysis stated that the non-treated sample displayed the most pronounced variability, which is contrary to the low PDI value (measured using the DLS technique) exhibited by the mentioned sample. However, for the presentation of TEM analysis results, representative images of liposome morphology were chosen, and they do not reflect the size of all particles in the sample. In addition, unlike microscopy, DLS cannot measure the size of individual particles within aggregates. As a result, the hydrodynamic radius values obtained from dynamic light scattering are often significantly larger, sometimes by an order of magnitude, than those observed using TEM. Since dynamic light scattering is inherently more responsive to larger particles, the scattering intensity it detects is proportional to the sixth power of the particle radius. Consequently, even though larger particles may be present at lower concentrations, they can dominate the scattering signal and obscure the detection of smaller particles that are more abundant [57]. There are various factors that may account for the differences in size distribution observed for liposomes when assessed by DLS versus TEM, including measurement mechanisms. Namely, DLS evaluates the hydrodynamic diameter by analyzing fluctuations in light scattering intensity, which reflects the particle’s overall motion in solution, including contributions from hydration layers and surface interactions. In contrast, TEM offers a direct image of the liposomes in a dehydrated or stained state, revealing their core structure without the influence of surrounding water molecules. TEM enables direct visualization of morphology, while DLS provides quantitative size distribution and stability assessment, ensuring a comprehensive understanding of their physical properties, uniformity, and size stability [58]. Hence, the sonicated population of liposomes showed a higher PDI (measured by DLS technique) due to the presence of aggregates (or larger vesicles), which was observed in the graphical presentation of DLS measurements (Appendix A) and was not visible in TEM images because of the above-mentioned facts. Notably, the sonicated liposome preparation showed characteristic traces (impurities or debris from the ultrasound probe) located in the central region of most observed liposomes.

### 3.7. The Radical Scavenging Activity of Developed Liposomal Formulations

The data related to the radical scavenging capacity of pure carob extract and developed liposomes are presented in Figure 6. The antioxidant capacity of the extract and liposomal formulations determined in the DPPH and ABTS assays is presented as the percentage of free radical neutralization.

The anti-DPPH activity of pure extract (diluted to achieve the same concentration as in the liposome samples) amounted to 69.1 ± 1.0% (Figure 6). The mentioned activity was not significantly different in non-treated and UV-irradiated liposomal formulations (69.6 ± 0.8% and 70.3 ± 0.7%, respectively), indicating the ability of the prepared liposomal vesicles to retain anti-DPPH radical activity via encapsulation. The developed liposomes can retain a considerable level of the antioxidant effect of carob extract, which was also the case with the antioxidant activity of unencapsulated betel extract and its encapsulated counterpart [28]. Additionally, anti-DPPH capacity was significantly lower in sonicated liposomes (67.9 ± 0.4%). In our preliminary study [59], shortened sonication time (15 min) and lower amplitude (40%) provided the liposomes with higher anti-DPPH potential (69.4 ± 0.6%) in comparison to liposomes after prolonged exposure time and higher amplitude obtained in the present study. The antioxidant activity of carob extract determined in the ABTS test was 25.6 ± 0.6%, while the antioxidant potential of extract-loaded multilamellar vesicles was significantly higher, 33.8 ± 0.2%. The mentioned activity was significantly lower upon UV light treatment (31.2 ± 0.4%) and sonication (31.4 ± 0.7%). Noudoost et al. study [60] also demonstrated that green tea encapsulated in a liposomal carrier provided a higher antioxidant effect in comparison to the free extract. Results of the Jahanfar et al. study [11] demonstrated that the inhibition percentage of rosemary extract-loaded glycerosomes was higher compared to pure extract. The modified antioxidant potential of plant extracts during their entrapment in liposomal vesicles was expected due to new physicochemical properties and consequently changed biological activity of the complex (liposome bilayer–extract compounds), depending on the structure, diameter, as well as zeta potential of the formed liposomes. The data of Cortie and Else’s research [61] suggest that non-peroxidizable phospholipid components can exert an *antioxidant*-like action in membranes, and the mentioned potential can augment the action of traditional antioxidant compounds. The antioxidant potential of empty liposomes originating from antioxidants added to the initial phospholipid mixture has already been published [21], which can be the reason for the higher antioxidant capacity of carob extract-loaded liposomes compared to free extract. A significant drop in the antioxidant potential of sonicated liposomes was expected due to the possibility of ultrasound waves generating free radicals due to the extreme conditions, such as the vigorous collapse of cavitation bubbles and the sonolysis of water and other medium components [62], and consequently decreasing the antioxidant effect of developed liposomes. Sonicated liposomes with carob extract showed a lower antioxidant effect, also due to a lower encapsulation efficiency, and thus, their non-encapsulated carob extract compounds can undergo destruction caused by environmental conditions, which can later result in a decrease in antioxidant capacity [28]. Also, as mentioned earlier, UV irradiation can trigger ROS damage in the liposomal bilayer [32], decreasing the antioxidant potential of the developed liposomal preparation.

### 3.8. Carob Extract Polyphenol Release Under Simulated Gastrointestinal Conditions

The carob extract polyphenols’ release from the pure extract and the encapsulation systems (non-treated, UV-irradiated, and sonicated liposomes) toward SGF (pH~1.5) and SIF (pH~7.4) were performed, and the release profiles are displayed in Figure 7A and Figure 7B, respectively. Results obtained from the polyphenol release studies of carob extract and carob extract-loaded liposomes were analyzed to examine the diffusion coefficients and diffusion resistances derived from liposomes in simulated gastric and intestinal fluids and the calculation flow is presented in the Appendix A. The data on the diffusion coefficients and diffusion resistances are presented in Table 2. UV spectra of carob polyphenols in SGF and SIF medium are presented in Appendix A.

As can be seen from Figure 7A, the release of polyphenols within SGF from the pure extract and sonicated liposomes was faster and higher compared to larger-sized non-treated, and UV-irradiated liposomal vesicles. Namely, the amount of released polyphenol compounds after 3 h is 33.14% from the pure extract and 29.91% from sonicated liposomes, while the level of distributed polyphenols from non-treated and UV-irradiated liposomes in the SGF medium was 11.22% and 12.37%, respectively. Diffusion coefficients of non-treated and UV-irradiated liposomes in the SGF medium were similar and amounted to 9.44 × 10^−9^ and 9.77 × 10^−9^ m^2^/s, respectively, while the mentioned variable was higher for other tested samples: 1.61 × 10^−8^ m^2^/s for pure extract and 1.65 × 10^−8^ m^2^/s for sonicated liposomes (Table 2). Consequently, diffusion resistance in the SGF surrounding follows the same trend: 2.54 × 10^5^ s/m (extract), 4.32 × 10^5^ s/m (non-treated), 4.17 × 10^5^ s/m (UV-irradiated), and 2.47 × 10^5^ s/m (sonicated) (Table 2). A slightly higher diffusion coefficient, as well as a lower diffusion resistance value of UV-treated liposomes in comparison to non-treated parallels, can be explained by the potential creation of pores within liposomal membranes induced by UV irradiation [63]. Several studies have shown the higher release of encapsulated active compounds from liposome particles upon photoinitiated destabilization using UV lights [32,64,65]. The reason for the lower capacity of sonicated liposomes to hold carob polyphenols lies in a greater contact surface between smaller liposomal particles and the simulated fluids. In the case of small liposomes, such as those after sonication, the contact surface with the surrounding medium increases. Hence, small particles (a high number of particles), as well as high-porosity samples, are more exposed to the surrounding medium influences due to an increase in the available areas for surface contact [66]. Additionally, the significantly lower viscosity of the sonicated liposomal system compared to other liposomes (shown in Table 1) can also be responsible for a higher percentage of released polyphenols in the environmental medium. As the degree of retention of carob polyphenols in liposomes was high in SGF (88.78% for non-treated and 87.62% for UV-irradiated), the data demonstrated the potential of liposomal vesicles to retain and protect entrapped bioactives in the presence of pepsin and low pH value in the gastric surrounding, which agrees with the literature [67]. On the other hand, enzymes present in the SIF medium, including lipase and phospholipase A2, as well as bile salts, may destabilize the liposomal bilayer membrane due to the catalysis of lipid hydrolysis [24], resulting in a more rapid and more abundant release of carob phenolics, which is presented in the next paragraph. Namely, the presence and absence of enzymes or bile salts in the investigated medium for the release kinetics, as well as medium pH, significantly affect the release properties of the liposomal carriers [24,67].

It can be observed that the polyphenol release from the pure extract was faster and complete in the SIF medium (94.7% after 8 h, Figure 7B), while all liposomal particles allowed a more retarded release of encapsulated extract bioactives: 54.0% for non-treated, 51.9% for UV-irradiated, and 61.5% for sonicated liposomes. The reason for the lower level of distributed carob polyphenols from all liposomal systems obtained in the SIF medium, i.e., their incomplete recovery, is in the rigid liposomal membrane. Namely, in this study, only phospholipid compounds (in the absence of sterols) were used for the preparation of liposomes. According to the references, the addition of sterols during liposome formation can change the fluidity and permeability of membranes [41,68]. Nevertheless, cholesterol can affect the mechanical properties of membranes due to modifications of the acyl chain order, as well as the interfacial membrane region, but the mentioned effects are not universal and depend on the type of present lipids and cholesterol concentration [68,69]. At low and intermediate percentages, cholesterol enhances the membrane fluidity in the upper region [68]. Therefore, more fluid membranes are able to release a higher amount of encapsulated components, but at the same time, their undesirable leakage during the storage period. Furthermore, cholesterol, as a commonly used sterol in liposome systems, can exert negative effects on health as well. Since the effects of developed liposomes with carob extracts on the metabolism of rat models will be included in future experiments, plant-based sterols, such as β-sitosterol, and fungus-based sterols, such as ergosterol, can be taken into account to improve the recovery of polyphenols from liposomal vesicles. Also, the liposome bilayer membrane containing β-sitosterol shows a higher level of fluidity in comparison to cholesterol parallels [41]. Furthermore, diffusion coefficients determined in the SIF were higher compared to the values obtained for SGF: 3.14 × 10^−8^ m^2^/s (extract), 1.90 × 10^−8^ m^2^/s (non-treated), 1.35 × 10^−8^ m^2^/s (UV-irradiated), and 2.07 × 10^−8^ m^2^/s (sonicated), while diffusion resistance values were 1.29 × 10^5^ s/m (extract), 2.14 × 10^5^ s/m (non-treated), 3.01 × 10^5^ s/m (UV-irradiated), and 1.97 × 10^5^ s/m (sonicated) (Table 2). Although the diffusion coefficient for UV-irradiated liposomes was lower in comparison to the non-treated parallel, the percentage of distributed carob polyphenols was the same after the 8 h period in the SIF medium. At the beginning of the diffusion process, the access of enzymes and salts from the SIF medium to liposomal membranes was probably slower and more difficult due to larger particles and/or formed agglomerates in the UV-irradiated sample, causing slow diffusion, i.e., higher diffusion resistance. On the other hand, the diffusion coefficient and the level of distributed polyphenols from sonicated parallel in SIF were higher due to a higher surface area between the liposome bilayer and medium enzymes, including lipases, and bile salts, as well as reduced viscosity, as mentioned above. Shashidhar and Manohar’s study [70] reported that a higher viscosity liposome system provided a slower release of encapsulated active principles. Namely, in the case of reduced viscosity, diffusion and mass transfer of molecules are improved, which consequently provides their faster and more intensive release into the surrounding medium.

Due to the shown results of carob polyphenol release in simulated gastrointestinal conditions and similar conclusions obtained in several studies [24,37,67], liposomal encapsulation of herbal extract formulations and plant-based ingredients can be proposed for increased gastric stability and delayed/targeted recovery of plant bioactives in the intestine.

Although carob pulp extracts encapsulated in liposomes demonstrate significant potential, it is crucial to advance toward pharmacological validation of their health-promoting activities. The encapsulation of carob pulp extracts in liposomes holds significant promise for enhancing their bioactive properties, particularly antioxidant and anti-inflammatory effects, which could be utilized effectively in therapeutic applications, such as reducing oxidative stress and inflammation or preventing cellular damage in various disease models [71]. Advanced in vivo studies could investigate their biodistribution, pharmacokinetics, and targeted release mechanisms, paving the way for novel treatments, dietary supplements, and functional foods designed to optimize health benefits. Furthermore, the potential of liposome-encapsulated carob extracts to address specific conditions, like gut inflammation, vascular oxidative damage, or mitigating toxic treatment side effects, could add to their innovative applications in both pharmacology and nutraceuticals.

## 4. Conclusions

In this study, the physicochemical characteristics of *C. siliqua* (carob) pulp flour extract-loaded liposomes are evaluated. The vesicle size and PDI results indicated the existence of a multilamellar and uniform liposomal system before post-formulation treatment. The encapsulation efficiency was high; however, medium values of the zeta potential and an increase in liposome size and heterogeneity of developed liposomes suggest that future experiments should focus on improving their stability. NTA showed that the sonicated liposomes demonstrated a higher concentration of vesicles in comparison to two other liposomal populations, whose concentrations did not differ. FT-IR spectra confirmed the formation of low-stability oxygen species after UV irradiation and sonication of liposomes, which is presumably the cause of changes in the structure of the liposomal membrane and the consequent leakage of carob polyphenols. TEM analysis revealed purified liposomal vesicles with preserved structural integrity before and after modifications, while all samples showed size heterogeneity. The polyphenol distribution in the simulated gastric conditions from the pure extract and sonicated liposomes was faster and higher in comparison to non-treated and UV-irradiated liposomes, but still lower than in the presence of intestinal enzymes and bile salts. The polyphenol release from the pure extract was faster and complete in the simulated intestinal conditions, whereas liposomal formulations provided a more controlled/prolonged release. Our results also showed that carob pulp extract, as well as its different liposomal nutraceutical formulations, show in vitro antioxidant potential that may be of interest for the further development of functional food in order to improve the prevention and treatment of various diseases whose pathogenesis is based on oxidative stress.

## Figures and Tables

**Figure 1 pharmaceutics-17-00776-f001:**
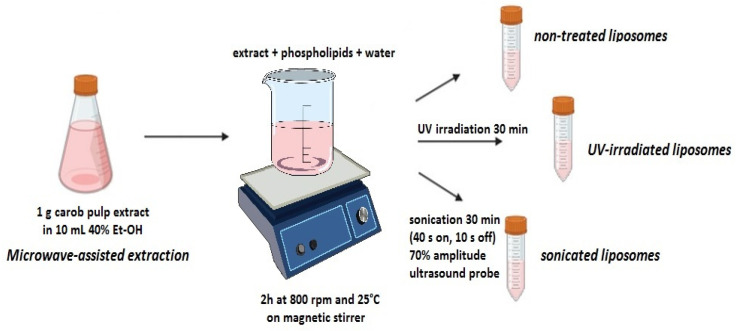
Principal scheme of the carob extract-loaded liposome preparation and modification methods.

**Figure 2 pharmaceutics-17-00776-f002:**
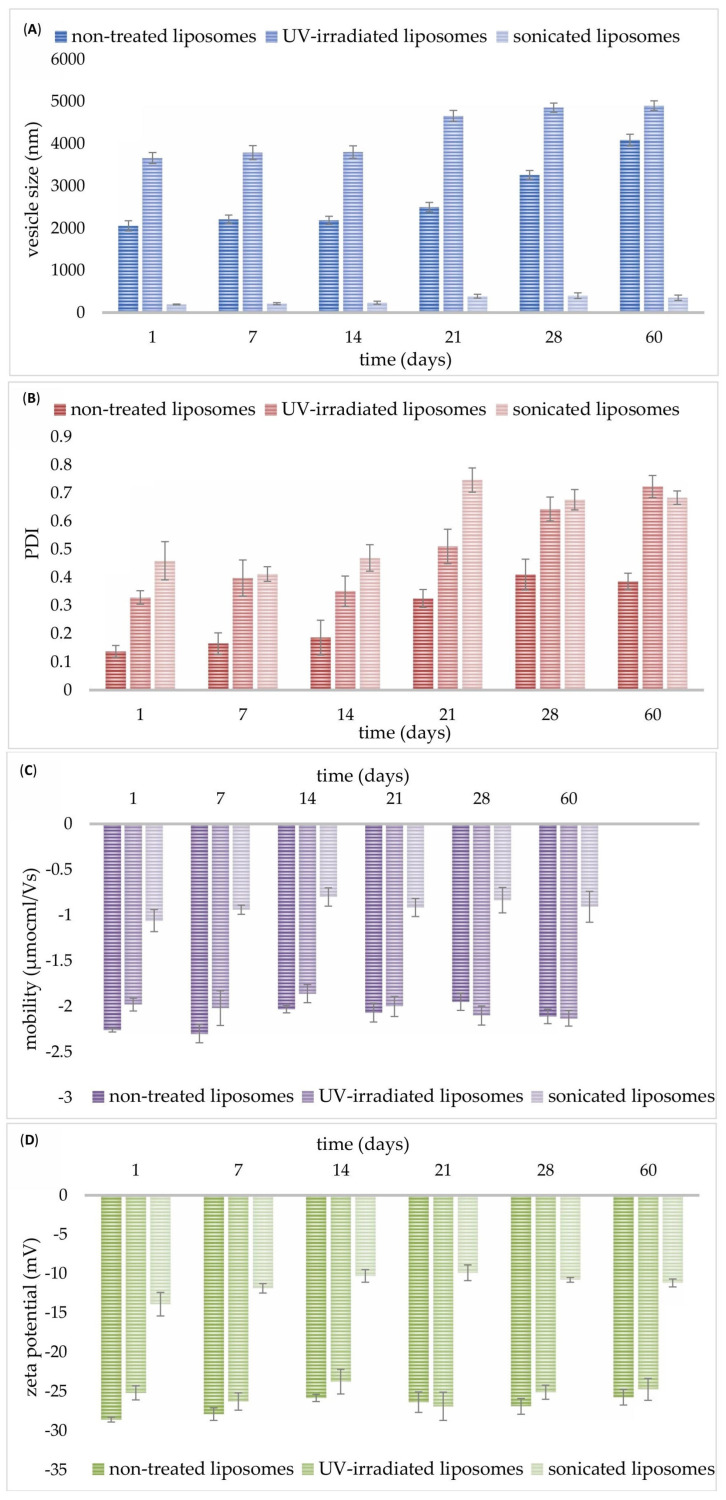
Storage stability of developed carob extract-loaded liposomes (non-treated, UV-irradiated, and sonicated samples): (**A**) vesicle size, (**B**) polydispersity index, (**C**) mobility, and (**D**) zeta potential monitored during 60 days of storage at 4 °C.

**Figure 3 pharmaceutics-17-00776-f003:**
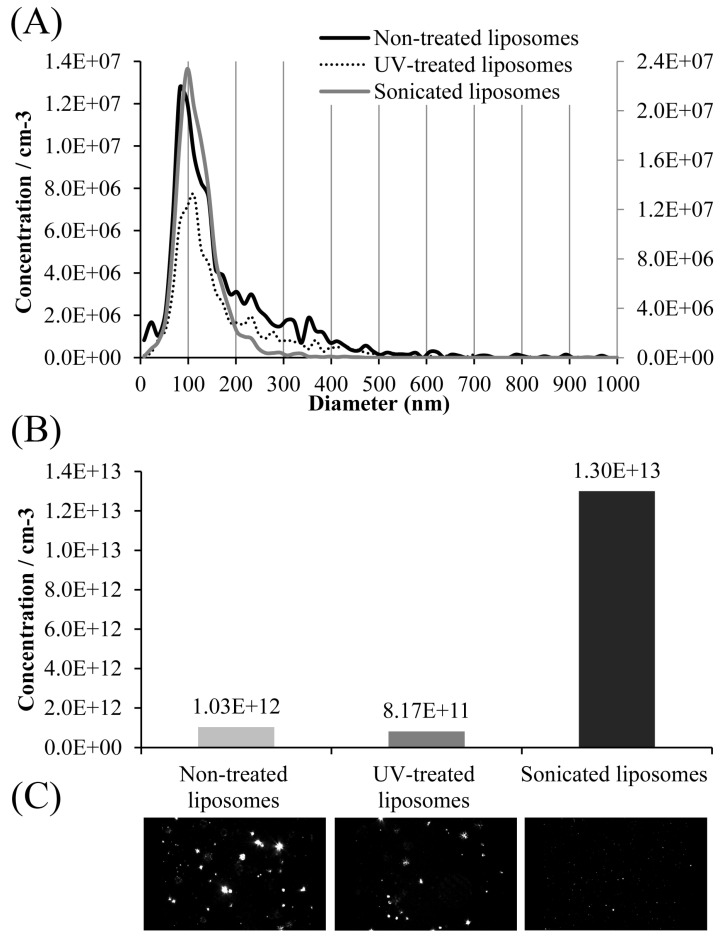
Size distribution (**A**), concentration (**B**), and representative video frame capture (**C**) of non-treated, UV-irradiated, and sonicated carob extract-loaded liposomes. Data on size distribution and concentration represent the mean values of three independent measurements obtained by nanoparticle tracking analysis.

**Figure 4 pharmaceutics-17-00776-f004:**
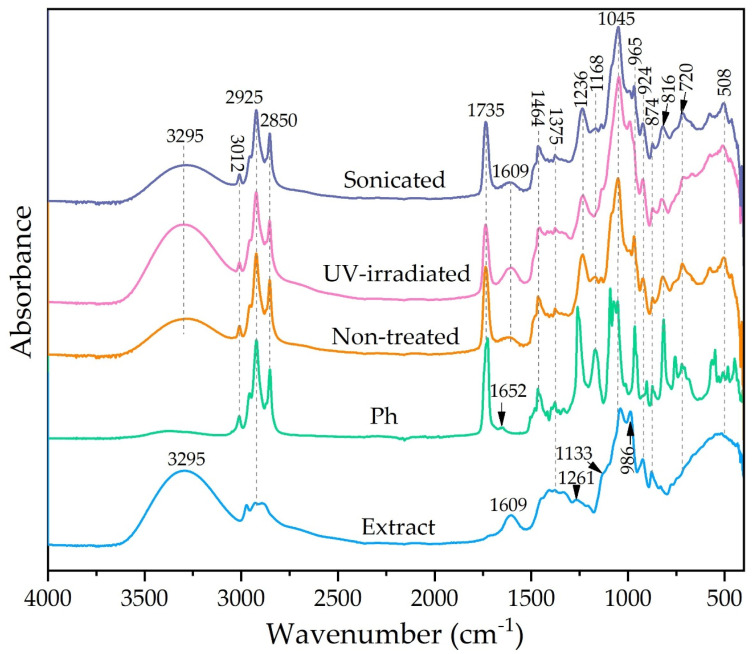
FT-IR spectroscopic analysis of pure carob extract, phospholipids (Ph), and developed carob extract-loaded liposomes (non-treated, UV-irradiated, and sonicated samples).

**Figure 5 pharmaceutics-17-00776-f005:**
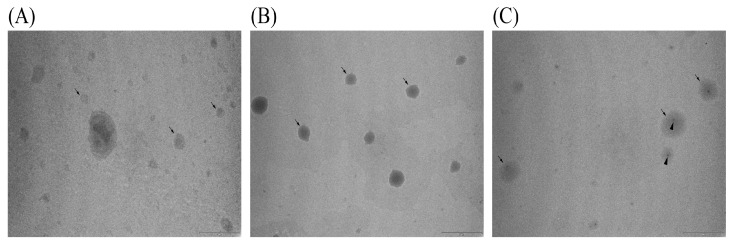
Representative transmission electron microscopy images showing non-treated (**A**), UV-irradiated (**B**), and sonicated (**C**) carob extract-loaded liposomes. Bar—1 µm. The arrow represents the common appearance of liposomes in a corresponding sample. The triangle represents impurities or debris from the ultrasound probe within the liposomes.

**Figure 6 pharmaceutics-17-00776-f006:**
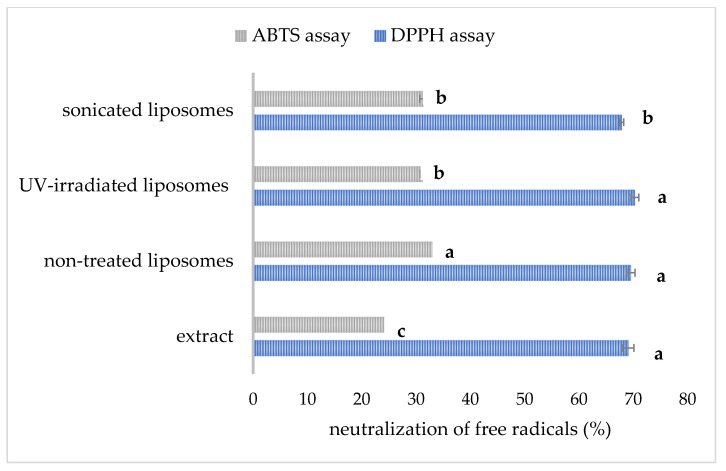
The radical scavenging capacity of pure carob extract and developed carob extract-loaded liposomes (non-treated, UV-irradiated, and sonicated samples); different letters (a–c, for both assays separately) indicate that there was a statistically significant difference based on Duncan’s post hoc test at *p* < 0.05 level, n = 3, mean value ± standard deviation.

**Figure 7 pharmaceutics-17-00776-f007:**
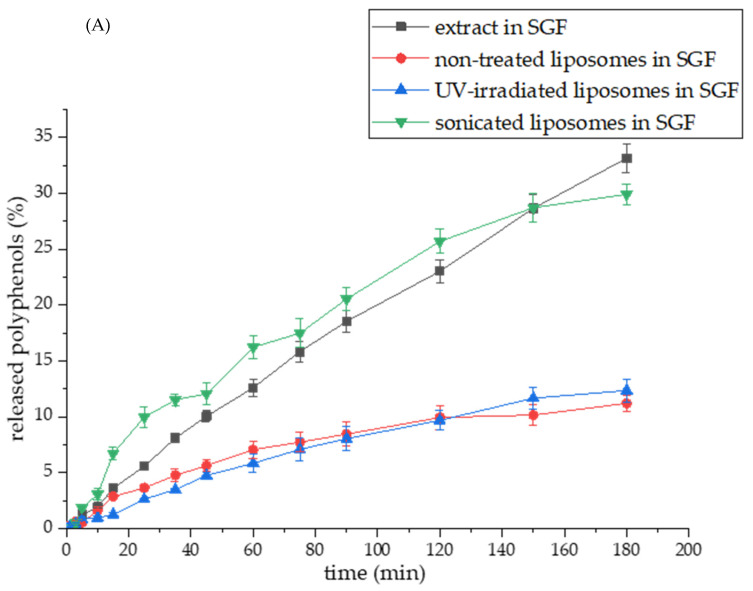
Release profiles of polyphenols from carob extract and developed carob extract-loaded liposomes (non-treated, UV-irradiated, and sonicated samples) in (**A**) simulated gastric fluid (SGF, pH 1.5, monitored for 3 h) and (**B**) simulated intestinal fluid (SIF, pH 7.4, monitored for 8 h) at 37 °C.

**Table 1 pharmaceutics-17-00776-t001:** Encapsulation efficiency and rheological properties of developed liposomal formulations with carob extract determined on the 1st and 60th days of storage.

Day	Liposomes	Encapsulation Efficiency (%)	Viscosity (mPa·s)	Surface Tension (mN/m)	Density (g/cm^3^)
1st	Non-treated	80.59 ± 1.29 ^a^*	18.40 ± 1.22 ^a^	41.75 ± 1.98 ^a^	1.04 ± 0.01 ^a^
	UV-irradiated	74.99 ± 1.02 ^b^	16.90 ± 0.85 ^a^	39.57 ± 1.06 ^a^	1.04 ± 0.02 ^a^
	Sonicated	71.05 ± 1.34 ^c^	5.17 ± 0.09 ^c^	24.83 ± 1.27 ^c^	1.02 ± 0.03 ^b^
60th	Non-treated	78.84 ± 1.95 ^a^*	13.93 ± 0.82 ^b^	30.20 ± 0.93 ^b^	1.04 ± 0.00 ^a^
	UV-irradiated	69.51 ± 0.91 ^c^	14.20 ± 1.03 ^b^	29.27 ± 1.00 ^b^	1.05 ± 0.01 ^a^
	Sonicated	48.62 ± 1.78 ^d^	4.46 ± 0.14 ^d^	22.03 ± 0.94 ^d^	0.99 ± 0.02 ^b^

* Different letters in each column refer to statistically significant differences among various samples (for each variable separately) regarding the results of statistical analysis in one-way analysis of variance and Duncan’s post hoc test at *p* < 0.05 (n = 3).

**Table 2 pharmaceutics-17-00776-t002:** Diffusion coefficients (D) and diffusion resistance (R) of the carob extract and carob extract-loaded liposomes (non-treated, UV-irradiated, and sonicated samples) in simulated gastric fluid (SGF) and simulated intestinal fluid (SIF).

Medium	Samples	D [m^2^/s]	R [s/m]
SGF	Extract	1.61 × 10^−8^	2.54 × 10^5^
	Non-treated liposomes	9.44 × 10^−9^	4.32 × 10^5^
	UV-irradiated liposomes	9.77 × 10^−9^	4.17 × 10^5^
	Sonicated liposomes	1.65 × 10^−8^	2.47 × 10^5^
SIF	Extract	3.14 × 10^−8^	1.29 × 10^5^
	Non-treated liposomes	1.90 × 10^−8^	2.14 × 10^5^
	UV-irradiated liposomes	1.35 × 10^−8^	3.01 × 10^5^
	Sonicated liposomes	2.06 × 10^−8^	1.97 × 10^5^

## Data Availability

The datasets generated during and/or analyzed during the current study are available from the corresponding author upon reasonable request.

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
