# Peer review of "Liposomal Encapsulation of Carob (Ceratonia siliqua L.) Pulp Extract: Design, Characterization, and Controlled Release Assessment†"

_pharmaceutics, 2025, doi:10.3390/pharmaceutics17060776_

Round 1

Reviewer 1 Report

Comments and Suggestions for Authors

The current study aimed to encapsulate carob extract into liposomal formula, the study is interesting, however, the following should be addressed;

  • The amount of phospholipid (4 g), volume of extract used, and volume of hydration phase, on what basis was this amount determined?
  • Please provide a detailed section on analytical quantification of polyphenols in the extract.
  • Provide TEM for untreated, UV-irradiated, and sonicated formulae to track morphological alterations.
  • Enhance the resolution of figures 1&3
  • What were the conditions of the storage stability study?

Author Response

Dear Reviewer,

The authors thank you for your time and efforts that improved the manuscript. We have thoroughly revised our manuscript, taking into account all your recommendations, and greatly appreciate all comments and suggestions because their implementation will significantly improve the manuscript. We hope that the manuscript has been improved and is acceptable for publication.

Sincerely Yours,

Authors

Reviewer #1

Reviewer: The current study aimed to encapsulate carob extract into liposomal formula, the study is interesting, however, the following should be addressed;

The amount of phospholipid (4 g), volume of extract used, and volume of hydration phase, on what basis was this amount determined?

Authors: The ratio of phospholipids and ultrapure water was determined in our previous study that dealt with rosehip extract-loaded liposomes (https://doi.org/10.3390/plants12173063). In addition, the amount of added carob extract in a final liposomal formulation was determined in a preliminary screening, and the selection was made based on the highest encapsulation efficiency. Due to the Reviewer's kind query, additional explanation was provided in Section 2.3. All changes are highlighted in red in the manuscript.

Reviewer: Please provide a detailed section on analytical quantification of polyphenols in the extract.

Authors: We thank the Reviewer for this suggestion. We would like to clarify that the detailed analytical quantification of polyphenols in the carob extract used, as well as the chromatogram figure, were previously published in our earlier studies (https://doi.org/10.3390/pharmaceutics14030657, https://doi.org/10.3390/separations10090465), which are now clearly cited in the revised manuscript. Nonetheless, to ensure clarity and completeness in the current work, we have expanded Section 2.2 to summarize the key elements of the HPLC-DAD method used. This includes the detection wavelengths for different phenolic groups, the use of external standards, and the presentation of quantitative results. The concentrations of the seven identified phenolic compounds are reported with standard deviations, and limits of detection are also included for compounds not detected.

Reviewer: Provide TEM for untreated, UV-irradiated, and sonicated formulae to track morphological alterations.

Authors: Thank you very much for your suggestion. TEM analysis was performed, and the obtained figures and results were provided in the revised version of the manuscript (Figure 5, Section 3.6).

Reviewer: Enhance the resolution of figures 1&3

Authors: We are very grateful that you pointed out this omission to us. The resolution of both figures was improved.

Reviewer: What were the conditions of the storage stability study?

Authors: Due to the Reviewer's kind suggestion, the conditions used in the storage stability study were provided in Section 2.6, as well as in the titles of figures.

Reviewer 2 Report

Comments and Suggestions for Authors Dear Authors, Thank you very much for the submission of the manuscript entitled "Liposomal Encapsulation of Carob (Ceratonia siliqua L.) Pulp Extract: Design, Characterization, and Controlled Release Assessment". The study seems to be interesting, well-structured and has practical significance. I have only several minor comments and suggestions. Please, find my recommendations below: 1. P. 3, subsection 2.3. Development and modification of carob extract-loaded liposomes I kindly recommend the Authors adding a principal scheme of the liposomes preparation method. 2. P. 3, subsection 2.4. Please, provide the equation for the encapsulation efficiency calculation. At the same time, drug loading capacity should be calculated. 3. P. 4, line 188; P. 5, line 198 I kindly recommend using the equation editor for the formulas. 4. P. 5, line 203 Release kinetics mathematical models should be used for data analysis and release mechanisms understanding. 5. P. 18, Figure 4 Please, demonstrate pH values for Figure A and B. Also, labels "A" and "B" should be added into figures. 6. Did the Authors analyse thermal properties by TGA and DSC techniques?   Additional suggestion Please, use the MDPI format and style for text and subsections titles.

Author Response

Dear Reviewer,

The authors would like to thank you for your time and efforts that improved the manuscript. We have thoroughly revised our manuscript, considered all your recommendations, and greatly appreciate all comments and suggestions because their implementation will significantly improve the manuscript. We hope that the manuscript has been improved and is acceptable for publication.

Sincerely Yours,

Authors

Reviewer #2

Reviewer: Dear Authors,

Thank you very much for the submission of the manuscript entitled "Liposomal Encapsulation of Carob (Ceratonia siliqua L.) Pulp Extract: Design, Characterization, and Controlled Release Assessment". The study seems to be interesting, well-structured and has practical significance. I have only several minor comments and suggestions. Please, find my recommendations below: 1. P. 3, subsection 2.3. Development and modification of carob extract-loaded liposomes I kindly recommend the Authors adding a principal scheme of the liposomes preparation method.

Authors: Due to the Reviewer’s kind suggestion, a principal scheme of the liposomes preparation and modification methods was provided as Figure 1.

Reviewer: 2. P. 3, subsection 2.4. Please, provide the equation for the encapsulation efficiency calculation. At the same time, drug loading capacity should be calculated.

Authors: Regarding the Reviewer's valuable query, the equation for the encapsulation efficiency calculation was provided in section 2.4, and all changes are highlighted using track changes in the whole manuscript. However, in the case of carob extract-loaded liposomes, the drug loading capacity was not calculated. Namely, we used the whole extract, which is not standardized, i.e., the amounts of target compounds varied depending on the series of carob pulp flour used; thus, we do not have the information related to the weight of the main compounds in the extract, necessary for the calculation of drug loading capacity for all target compounds. Due to the Reviewer’s kind suggestion, future experiments will include the HPLC analysis of prepared liposomes with carob extract, as well as the standardization of the extract sample, and thus, the calculation of drug loading capacity for all target compounds.

Reviewer: 3. P. 4, line 188; P. 5, line 198 I kindly recommend using the equation editor for the formulas.

Authors: Thank you very much for your significant notice. We used the equation editor for all formulas (the two mentioned and one newly added for the encapsulation efficiency).

Reviewer: 4. P. 5, line 203 Release kinetics mathematical models should be used for data analysis and release mechanisms understanding.

Authors: The calculation flow related to the diffusion of polyphenols from liposomes and extract, diffusion coefficients, and diffusion resistance with corresponding formulas and explanations, is presented in the supplementary material. This is now indicated in section 2.9.

Reviewer: 5. P. 18, Figure 4 Please, demonstrate pH values for Figure A and B. Also, labels "A" and "B" should be added into figures.

Authors: Due to the Reviewer’s kind suggestion, pH values are provided in the figure title, as well as "A" and "B" in the figures.

Reviewer: 6. Did the Authors analyse thermal properties by TGA and DSC techniques?  

Authors: We completely agree that thermal properties are very important for liposomal formulations and their further implementation. However, in this stage, we did not analyze the thermal characteristics of the developed liposomes with carob extract. Namely, the following step of our investigation will be the analysis of the biological potential of developed liposomes with carob extract (in the obtained form without exposure to higher temperatures or industrial conditions) in the animal model (temperature of ~37°C). In the case of satisfied and promising results, the thermal characteristics of liposomes with carob extracts will be determined using TGA analysis with the aim of optimizing conditions for potential future formulations. The DSC technique will not be included for the analysis of thermal properties of developed liposomal systems due to the facts already described in the literature and our previous study (https://doi.org/10.1002/ejlt.201800039). Namely, the melting of the fatty acyl chains of a single phospholipid type occurs at a well-defined temperature, whereas for a phospholipid mixture (used in the case of carob extract-loaded liposomes), the phase transition can be rather broad (https://doi.org/10.1016/0304-4157(76)90008-3, https://doi.org/10.1002/ejlt.201400481). For soy L-α-phosphatidylcholine, as a commercial mixture of phospholipids, it is not possible to detect phase transition in the temperature range from 0 to 100ËšC, using, available to us, the NANO DSC III instrument, because only pure phospholipids with saturated fatty acyl chains possess sharp phase transition, which could be detected in narrow temperature range.

Reviewer: Additional suggestion Please, use the MDPI format and style for text and subsections titles.

Authors: Thank you for your valuable suggestions and comments. Based on our experience, we have decided to merge the "Results" and "Discussion" sections to facilitate a smoother narrative flow and better connect different parts of the text related to various findings. We believe this adjustment enhances the coherence and readability of the paper. In the remainder of the manuscript, all information is presented according to the style and structure defined in the journal’s template, ensuring consistency with publication guidelines. We hope this revision improves the overall clarity of our research. Thank you for your time and constructive feedback.

Reviewer 3 Report

Comments and Suggestions for Authors

The study by Andrea Pirković et al. is devoted to the creation and improvement of classical systems for protecting active substances - liposomal delivery systems.

The work is well structured, but the number of references to sources is more surprising, the number of which is 100. I would recommend reducing the list of references to 30-50 sources, leaving only the most modern and fundamental ones.

In general, I liked the work, but I have a number of questions.

1) line 193 is a reference to the source in DOI format, it is necessary to assign the correct number

2) The biggest problem is the lack of an ESI section in the work. In which it is necessary to provide the initial data on DLS, tensiometry and viscometry.

2.1. What spindle and frequencies were used to change the rheology? Was the rheology detected with constant rotation or in oscillation mode?

2.2. What method was used to determine the surface tension, the ring or plate method? How many replicates were there?

2.3. Why was the NTA method not used to analyze the amount of liposomes formed in the solution?

2.4. In the experimental section it is written that the IR was recorded using the Smart iTR™ Attenuated Total Reflectance attachment (line 174 - 176), but in the work itself (Figure 2 line 555) the spectrum is given in transmission coordinates. This contradicts the physical principle by which the IR spectra were detected. The ATR spectrum is the spectrum of infrared light absorption by the sample.

3) There is no direct evidence that vesicles were formed. TEM images must be provided.

4) Individual spectra of all compounds in the esi section must be provided.

5) I agree that in the article itself it is possible to provide only kinetic curves, but for reliability in the ESI section it is necessary to provide the initial data in the form of UV spectra of polyphenols.

In the work itself there are practically no initial data/curves/spectra, which is why I have doubts about the reliability of the work.

Author Response

Dear Reviewer,

The authors thank you for your time and efforts that improved the manuscript. We have thoroughly revised our manuscript, considered all your recommendations, and greatly appreciate all comments and suggestions because their implementation will significantly improve the manuscript. We hope that the manuscript has been improved and is acceptable for publication.

Sincerely Yours,

Authors

Reviewer #3

Reviewer: The study by Andrea Pirković et al. is devoted to the creation and improvement of classical systems for protecting active substances - liposomal delivery systems.

The work is well structured, but the number of references to sources is more surprising, the number of which is 100. I would recommend reducing the list of references to 30-50 sources, leaving only the most modern and fundamental ones.

Authors: Due to the Reviewer’s kind suggestion, the number of references was significantly reduced by 35%.

Reviewer: In general, I liked the work, but I have a number of questions.

1) line 193 is a reference to the source in DOI format, it is necessary to assign the correct number

Authors: Thank you very much for your notice. We apologize for this. The correct number was assigned.

Reviewer: 2) The biggest problem is the lack of an ESI section in the work. In which it is necessary to provide the initial data on DLS, tensiometry and viscometry.

Authors: Due to the Reviewer’s kind suggestion, initial data on DLS, tensiometry, and viscometry measurements are provided in the supplementary material. Namely, row data on DLS measurements are provided as graphs in Figures 1S-3S, while viscometry and tensiometry measurements (presented as measurements in triplicate) are shown in Table 1S.

Reviewer: 2.1. What spindle and frequencies were used to change the rheology? Was the rheology detected with constant rotation or in oscillation mode?

Authors: Thank you very much for your questions. We provided all necessary information in the revised version of the manuscript, which will give future readers and researchers a better insight into how the measurement was performed. Specifically, the following information were included in the manuscript (section 2.5): non-treated, UV-irradiated, or sonicated liposomes in a volume of 6.7 mL were placed in a VOL-C-RTD chamber with VOLS-1 adapter and cylinder stainless steel spindle (single VOL-SP-6.7 spindle), and the measurements were done in three repetitions at 25°C on the 1st and 60th days. The viscosity was measured with constant rotation at 200 rpm, and the deflection, as a measurement of the torque, was >30 M%.

Reviewer: 2.2. What method was used to determine the surface tension, the ring or plate method? How many replicates were there?

Authors: Thank you for your query with the aim of providing all necessary information for future readers and researchers to have better insight into how the measurements were performed. Specifically, the following information related to surface tension, as well as density were included in the manuscript (section 2.5): The surface tension was determined using the Wilhelmy plate (range from 1 to 999 mN/m and resolution of 0.1 mN/m), while density was determined using a silicon crystal as the immersion body. The measurements were done in three repetitions for density and at least three repetitions for surface tension.

Reviewer: 2.3. Why was the NTA method not used to analyze the amount of liposomes formed in the solution?

Authors: Due to the Reviewer’s kind suggestion, the NTA method was used for the additional analysis of the developed liposomes, and the obtained data are provided in a revised version of the manuscript (Figure 3).

Reviewer: 2.4. In the experimental section it is written that the IR was recorded using the Smart iTR™ Attenuated Total Reflectance attachment (line 174 - 176), but in the work itself (Figure 2 line 555) the spectrum is given in transmission coordinates. This contradicts the physical principle by which the IR spectra were detected. The ATR spectrum is the spectrum of infrared light absorption by the sample.

Authors: The FTIR spectra were recorded using a Nicolet iS10 Fourier-Transform Infrared (FTIR) Spectrophotometer (Thermo Scientific, Waltham, USA) with small sample placement on ATR on a diamond. We know that in recording spectra in transmission mode, IR irradiation passes the total physical thickness of the sample, while in ATR recording, the pass of light is a subordinate of both the wavelength and the refractive index difference between the sample and the ATR crystal, giving infrared light absorption. After recording on the spectrophotometer, FTIR spectra were obtained in absorbance mode. However, FTIR spectra are usually shown in transmission mode, which can be seen in many other published works(https://doi.org/10.3390/pharmaceutics16081036; https://doi.org/10.3390/pharmaceutics16030322; https://doi.org/10.3390/pharmaceutics15112605; https://doi.org/10.1016/j.fochx.2022.100370; https://doi.org/10.3390/foods13101576).

Accordingly, we have converted the obtained ATR spectrum into transmission mode (%) with the help of OMNIC software, which has algorithms capable of converting the ATR spectrum.

In the Figure below, we show spectra in both absorbance and transmission modes.

Figure A. FT-IR spectroscopic analysis of pure carob extract, phospholipids (Ph), and developed carob extract-loaded liposomes (non-treated, UV-irradiated, and sonicated samples).

So, comparing spectra from both Figures, intensities, shape, and position of bands do not differ, and thus the following text was not changed.

Figure B. FT-IR spectroscopic analysis of pure carob extract, phospholipids (Ph), and developed carob extract-loaded liposomes (non-treated, UV-irradiated, and sonicated samples).

In line with the Reviewer's remark, we change the figure to a new one given in absorbance mode.

Reviewer: 3) There is no direct evidence that vesicles were formed. TEM images must be provided.

Authors: Due to the Reviewer's significant suggestion, TEM analysis was performed, and the obtained figures and results were provided in the revised version of the manuscript (Figure 5).

Reviewer: 4) Individual spectra of all compounds in the esi section must be provided.

Authors: We would like to clarify that the detailed analytical quantification of polyphenols in the carob extract used, as well as the chromatogram figure, were previously published in our earlier studies (https://doi.org/10.3390/pharmaceutics14030657, https://doi.org/10.3390/separations10090465), which are now clearly cited in the revised manuscript. Nonetheless, to ensure clarity and completeness in the current work, we have expanded Section 2.2 to summarize the key elements of the HPLC-DAD method used. This includes the detection wavelengths for different phenolic groups, the use of external standards, and the presentation of quantitative results. The concentrations of the seven identified phenolic compounds are reported with standard deviations, and limits of detection are also included for compounds not detected.

Reviewer: 5) I agree that in the article itself it is possible to provide only kinetic curves, but for reliability in the ESI section it is necessary to provide the initial data in the form of UV spectra of polyphenols.

Authors: Due to the Reviewer’s valuable suggestion, the initial data in the form of UV spectra of carob extract polyphenols in simulated gastric and intestinal fluids are provided in the supplementary material.

Reviewer: In the work itself there are practically no initial data/curves/spectra, which is why I have doubts about the reliability of the work.

Authors: We have thoroughly revised our manuscript, considered all your suggestions, and provided all requested data related to the DLS, viscometry, and tensiometry measurements, as well as UV spectra of carob polyphenols; thus, we strongly believe that this is proof of the work reliability.

Round 2

Reviewer 1 Report

Comments and Suggestions for Authors

1-The authors have made the requested corrections; however, it is uncommon to find an ultrasound probe debris in the TEM micrograph, the authors should mention the impact of this debris on the characteristics of the formula, especially since sonication is an integral part of the preparation of sonicated liposomes. Also the authors stated that irradiated vesicles possess larger PS than untreated and sonicated but by reviewing the TEM, the irradiated vesicles showed comparable size to sonicated vesicles.

2-Discuss the high PDI values exhibited by sonicated vesicles compared to untreated ones.

3- The authors mentioned that the PDI value for the non-treated sample was low (0.137±0.021); however, the TEM analysis stated that the non-treated sample displayed the most pronounced variability, which is contrary to the low PDI value exhibited by the untreated formula.

Author Response

Dear Reviewer,

The authors thank you for your time and efforts that improved the revised version of the manuscript. We have thoroughly revised the new version of the manuscript, taking into account all your recommendations, and greatly appreciate all suggestions because their implementation will significantly improve the manuscript. We hope that the manuscript has been improved and is acceptable for publication.

Sincerely Yours,

Authors

Reviewer #1

Reviewer: -The authors have made the requested corrections; however, it is uncommon to find an ultrasound probe debris in the TEM micrograph, the authors should mention the impact of this debris on the characteristics of the formula, especially since sonication is an integral part of the preparation of sonicated liposomes. Also the authors stated that irradiated vesicles possess larger PS than untreated and sonicated but by reviewing the TEM, the irradiated vesicles showed comparable size to sonicated vesicles.

Authors: Thank you very much for your queries. In the manuscript (section Results and Discussion), we provided the reference and sentence that points out the appearance of titanium particles as debris from the ultrasound probe in lipid formulations as the main disadvantage of the sonication treatment by the ultrasound probe (all newly added information, explanations, and references are highlighted in yellow). We have already explained the influence of the mentioned debris on the surface tension of the developed liposomal formulation. Due to the Reviewer's valuable suggestion, additional explanations and references related to the differences between dynamic light scattering and TEM results were added to the manuscript.

Reviewer: -Discuss the high PDI values exhibited by sonicated vesicles compared to untreated ones.

Authors: We thank the Reviewer for this suggestion. The explanation and discussion related to the high PDI values exhibited by sonicated vesicles compared to untreated ones were improved.

Reviewer: - The authors mentioned that the PDI value for the non-treated sample was low (0.137±0.021); however, the TEM analysis stated that the non-treated sample displayed the most pronounced variability, which is contrary to the low PDI value exhibited by the untreated formula.

Authors: Due to the Reviewer's significant noticing and kind suggestion, the section Discussion was significantly improved via the expansion by additional explanations and references.

Reviewer 3 Report

Comments and Suggestions for Authors

The authors have done a lot of work on the article. We have answered all your questions. I recommend accepting the article for publication.

A little remark. The size of the jackal (Figure 5) is usually given in the drawing itself and not in the caption to it.

Author Response

We sincerely appreciate Reviewer's  thoughtful review and the recognition  of our paper. We appreciate the meticulous attention to detail and careful corrections that have been invaluable in refining our work. Fig 5 scale bar are present in the image. We have included a better resolution image to improve visibility. Please find figure in PDF in attachment.Thank you for your time, effort, and insightful feedback that improved our manuscript.

Round 3

Reviewer 1 Report

Comments and Suggestions for Authors

The authors have addressed all the required  comments